# Spanish Validation of the Internet Gaming Disorder Scale–Short Form (IGDS9-SF): Prevalence and Relationship with Online Gambling and Quality of Life

**DOI:** 10.3390/ijerph17051562

**Published:** 2020-02-28

**Authors:** Marta Beranuy, Juan M. Machimbarrena, M. Asunción Vega-Osés, Xavier Carbonell, Mark D. Griffiths, Halley M. Pontes, Joaquín González-Cabrera

**Affiliations:** 1Faculty of Education, Universidad Internacional de la Rioja (UNIR), Avenida de la Paz, 137, 26006 Logroño, Spain; marta.beranuy@unir.net; 2Faculty of Psychology, University of the Basque Country (UPV/EHU), Avenida de Tolosa, 70, 20018 Donostia, Spain; juanmanuel.machimbarrena@ehu.eus; 3Faculty of Health Sciences, Universidad Pública de Navarra (UPNA), Calle Cataluña, s/n, 31006 Pamplona, Navarra, Spain; mvegaose@educacion.navarra.es; 4Faculty of Psychology, Education and Sport Blanquerna, Universitat Ramon Llull. Calle Císter, 34, 08022 Barcelona, Spain; xaviercs@blanquerna.url.edu; 5International Gaming Research Unit, Psychology Department, Nottingham Trent University, Nottingham NG11 8NS, UK; mark.griffiths@ntu.ac.uk; 6University of Tasmania, School of Psychological Sciences, Newnham Campus, Building O, Launceston TAS 7250, Australia; contactme@halleypontes.com; 7The International Cyberpsychology and Addictions Research Laboratory (iCARL), University of Tasmania, Launceston TAS 7250, Australia

**Keywords:** Internet Gaming Disorder, gaming disorder, gaming addiction, behavioral addiction, Internet Gaming Disorder Scale-Short Form

## Abstract

Online gaming is a very common form of leisure among adolescents and young people, although its excessive and/or compulsive use is associated with psychological impairments in a minority of gamers. The latest (fifth) edition of the Diagnostic and Statistical Manual of Mental Disorders (DSM-5, Section III) tentatively introduced Internet Gaming Disorder (IGD). Since then, a number of evaluation tools using the DSM-5 criteria have been developed, including the Internet Gaming Disorder Scale–Short Form (IGDS9-SF). The main objective of this study was to translate and adapt the IGDS9-SF into Spanish, as well as to obtain indicators relating to its validity and reliability. The Spanish version of four scales were administered: IGDS9-SF, Mobile Phone-Related Experiences Questionnaire (CERM), Online Gambling Disorder Questionnaire (OGD-Q), and KIDSCREEN-27. The sample comprised 535 Vocational Training students (mean age 18.35 years; SD±2.13; 78.5% males) who reported playing video games in the past 12 months. Confirmatory factor analysis yielded a one-dimensional model with a good fit while the reliability indicators were satisfactory. Findings indicated that 1.9% of gamers were classified with IGD (meeting five or more criteria for more than 12 months). Additionally, another 1.9% were considered gamers ‘at-risk’ because they endorsed four criteria. Positive and significant relationships were found between the IGDS9-SF, the CERM, and the OGD-Q. Participants classified with IGD had poorer health-related quality of life. In conclusion, the Spanish IGDS9-SF is a valid and reliable instrument to assess IGD according to the DSM-5.

## 1. Introduction

The way in which individuals interact with technology is constantly evolving. New behaviors have emerged, communication and leisure activities have changed, and new psychological problems arose. In the late 1990s, concerns about the addictive use of the internet [1,2] were discussed and, since then, the concept has been extensively studied and debated [3,4,5,6]. Although it has been addressed from multiple perspectives and researchers have used different terms, ‘internet addiction’ has been one of the most commonly used terms, along with ‘problematic internet use’ [7,8,9]. Early research focused on internet-related and mobile-related behavior in general terms. However, over the years, studies have focused on more specific uses. This approach has been defined as the move from general problematic internet use (GPIU) to specific problematic internet use (SPIU) (e.g., [5]). Consequently, research has especially focused on internet gaming [10,11,12], online gambling [13,14,15], online sex/cybersex [16,17], and social media use [18,19,20].

Among the aforementioned problems, ‘Gaming Disorder’ (GD) has recently been introduced in the nosological manuals (American Psychiatric Association [APA], and World Health Organization [WHO],) by being included as a disorder under the heading of addictive behaviors. The DSM-5 [21] places the ‘Internet Gaming Disorder’ (IGD) in Section III (disorders requiring further investigation) and the International Classification of Diseases 11th (ICD-11) [22] considers GD among non-substance addictions.

IGD is considered an addictive behavior that does not involve the ingestion of a psychoactive substance, and is mainly characterized by recurrent and persistent participation in online video games, leading to clinically significant distress [21]. The nine IGD criteria contain the characteristics indicated in the components model of addiction [23], including salience, mood modification, tolerance, withdrawal, the conflicts it generates (whether interpersonal and/or intrapersonal), and the risk of relapse. The theoretical overlap between the components model of addiction and the nine IGD criteria has been previously ascertained empirically in earlier studies [24]. Despite this definition, the use of different theoretical frameworks has created difficulties in the conceptualization of a problem with worrisome prevalence data and which also produces harmful effects on those who suffer from it, making it a possible public health problem [25].

The video game industry generates millions of Euros in revenue every year (1.530 million Euros in 2018, 12.6% more than the previous year) as video gaming is considered one of the main forms of leisure for many audiences across all stages of life. According to the Spanish Video Game Association [26], the total number of gamers in 2018 amounted to 16.8 million (41% female). Their age ranged from 6 to 64 years old, although the youngest (6–14 years) stand out, and they played for an average of 6.2 h per week. Studies on participation in video games indicate the highest prevalence among younger populations [27], which appears to be an at-risk population due to specific features associated with adolescence (i.e., being at a developmental stage where there is little thought given to the long-term consequences of their actions [28]) and membership of the Z Generation (i.e., born during the early years of the 21^st^ century comprising individuals who have never known a world without the internet and mobile phones [29]).

Adolescent studies indicate a prevalence of IGD ranging from 1.7% to 10% [4,30,31,32,33]. According to the meta-analytical study by Fam [27], the prevalence of IGD among male adolescents is 6.8%, and 1.3% among females. In international samples, and without an established age range, the percentage ranges from 0.7% to 15.6% [34] or slightly higher in Chinese studies (from 3.5% to 17% [35]).

In Spain (where the present study was carried out), a study with a school sample of 708 students reported 72.8% online gamers [36] of whom 8.3% met five or more of the nine criteria for IGD (86.44% were male). Another study conducted with Spanish-speaking online gamers reported a prevalence of 2.6% disordered gamers [37]. This same study found that 6.5% were “engaged gamers at high risk”, and 11.9% were "engaged gamers at low risk”, with the remainder classified as “regular” or “casual” gamers. In a clinical sample of 86 disordered adolescent gamers, it was found that 96.6% were male [38].

In terms of comorbidity, IGD has been associated with a wide spectrum of psychological problems including depression, anxiety, social phobias, poorer school performance, and sleep disorders [36,39,40,41,42]. In addition, studies have begun to appear comprising clinical samples demanding psychological treatment, which meet the criteria for the disorder [38,43]. In the study by Martín-Fernández et al. [38], all participants had diagnostic comorbidity, in accordance with other studies [44,45]. The most prevalent comorbidities with IGD were found to be depression, social anxiety, ADHD, and aggressive behaviors [46,47]. Additionally, it is also important to examine the relationship of IGD with other non-substance addictions (particularly gambling) because several studies have established common risk factors such as personality traits [48], levels of impulsivity and compulsivity [49], and similarities in the neurobiological functioning of patients with IGD and patients with pathological gambling [50]. It is relevant to relate IGD with online gambling disorder or other problematic behaviors such as problematic smartphone use, because these three behaviors occur via Information and Communications Technologies (ICTs) and which typically meet addiction criteria similar to IGD (i.e., salience, mood modification, tolerance, withdrawal, conflict, and relapse) when people present with a clinical problem. Currently, there are psychometric tests that evaluate online gambling disorder specifically in adolescent populations [14]. In addition, the most used device to play or connect to the internet is via smartphone. Although a decade ago, gaming consoles and computers were the only technological hardware available to play online, in recent years, the use of the smartphone has significantly increased [51]. Currently smartphones are used by 21% of almost 17 million players in Spain (only surpassed by gaming consoles; 26%). This percentage increases considerably (up to 40%) in the 15–24-year age range. It should also be noted that in the Spanish context, the average time spent weekly with mobile gaming is 5.1 h, compared to 3.9 h on gaming consoles and the 4.9 h on computers [26]).

Many studies have focused on the negative effects of IGD on psychological health and other health-related problems, but fewer studies have linked it to more global general well-being constructs (such as health-related quality of life [HRQoL]). HRQoL is defined as a state of complete physical, mental, and social well-being that is perceived by individuals and by those around them (for more information see the review by Wallander and Koot [52]). The evaluation of HRQoL is complex because it is a polyhedral construct that presents multiple conceptual approaches although one of the best approaches for examining the infant-juvenile stage is with the KIDSCREEN [53], a psychometric test adapted in almost 30 languages. The HRQoL provides general indicators on the impact of a problem in areas relevant to an adolescent’s life, such as physical and psychological well-being or the relationship with parents and peers. Several studies indicate that inadequate use of the internet is related to low scores on HRQoL, in addition to a lower self-perceived social support and more friends only known through the internet [54,55]. Similarly, other studies indicate that both HRQoL and social aspects are affected among people presenting problems related to the use of video games in adolescence [31,36]. 

The APA’s [21] operationalization of IGD has arguably reduced the diversity in the assessment instruments as well as the number of items contained in each of these instruments, providing more uniform measures with high internal consistency and adequate criterion validity. A systematic review [56] indicated the nine-item Internet Gaming Disorder Scale-Sort Form (IGDS9-SF) [57] as the most reviewed and translated instrument to assess IGD based on the nine IGD criteria developed by the APA. In fact, since its publication, the IGDS9-SF has been translated into at least nine languages: Chinese [58], German [59], Czech [60], Slovenian [61], Italian [62], Persian [63], Turkish [64,65], Polish [66], European and South American Portuguese [67,68]. Therefore, it is a cross-culturally suitable psychometric tool to assess IGD that allows framing the problem uniformly and inter-culturally with adequate psychometric properties.

This IGDS9-SF is based on the nine criteria suggested by the DSM-5 for IGD that includes: (i) preoccupation with gaming; (ii) withdrawal symptoms; (iii) tolerance; (iv) unsuccessful attempts to reduce or quit gaming; (v) loss of interest in previous activities or entertainments as a result of (and with the exception of) gaming; (vi) continuing to game despite knowing the associated psychosocial problems; (vii) deceiving family members, therapists, or others about the amount of time spent on gaming; (viii) playing video games to evade or relieve negative moods; and (ix) jeopardizing or losing a meaningful relationship, job, or educational or employment opportunity due to gaming.

As aforementioned, the use of different theoretical frameworks to assess IGD generated problems in conceptualization and, given the broad international acceptance of the IGDS9-SF and its robust psychometric properties, the objective of the present study was to extend the cross-cultural psychometric assessment evidence-base related to the assessment of IGD by translating and adapting the IGDS9-SF into Spanish to ascertain its psychometric suitability to this specific cultural context in terms of the validity and reliability of its scores. The validation in this cultural context for the Spanish language is a priority, given that Spanish is the third most spoken language in the world (534 million speakers [69,70]) and there are 21 countries where Spanish is the official language [70]. Consequently, a Spanish version of the most used IGD assessment tool is needed to encourage and improve research investigating IGD in Spanish-speaking countries and to facilitate cross-culturally unified research of this emerging public health issue.

The secondary objectives of this study were to: (a) to obtain indicators of validity and reliability of the Spanish version of IGDS9-SF, including the confirmatory study of its factor structure; (b) to test whether the newly developed psychometric test works equally in both men and women, as well as in adolescences and young adults; (c) establish the prevalence of IGD in a sample of adolescents and young Vocational Training (VT) students aged between 15 and 25 years; and (d) examine the relationship between the IGD and HRQoL. To achieve the aforementioned objective, it was hypothesized that: (i) the IGDS9-SF would show adequate psychometric properties in the sample recruited, similarly to previous IGDS9-SF psychometric validation studies conducted in different countries [58,59,61,67]; (ii) the measurement model would be invariant across both genders [57]; (iii) the prevalence of IGD would be between 2% and 4%, which is referred to in other national and international studies [27,37]; and (iv) those who met the IGD criteria within the sample recruited would present lower scores on the different HRQoL dimensions [31].

## 2. Materials and Methods 

### 2.1. Design and Participants

The instrument validation study was conducted from February to May 2019. The sample was recruited from 17 VT centers in the Autonomous Community of Navarre by means of non-parametric incidental sampling. The distribution of students by cycles and school stages was as follows: basic VT (152 first-grade students, 14.2%; 70 second-grade students, 6.5%); middle degree VT (433 first-grade students, 40.4%; 56 second-grade students, 5.2%); and higher degree VT (325 first-grade students, 30.3%; 35 second-grade students, 3.4%). The initial number of participants was 1064 (593 males and 471 females), of whom 535 reported playing video games in the last 12 months. Of this final sample, 420 were males (78.5%) and 115 were females (21.5%). The mean and standard deviation of age was 18.35 years (±2.13), with a range of 15–25 years.

### 2.2. Instruments

The participants provided information about demographic variables including gender, grade, school, and age. They also indicated the name of the video game they spent the most hours on in the past 12 months and whether or not they considered themselves addicted to online video games. In addition, they completed the following assessment tools.

Spanish translation of the IGDS9-SF [57,61] (see the Spanish version in Appendix A). The IGDS9-SF assesses the severity of IGD and its detrimental effects, examining online and/or offline gaming activities that occur over a 12-month period with nine questions based on the DSM-5 IGD criteria that are rated on a five-point Likert scale: 1 (never), 2 (rarely), 3 (sometimes), 4 (often), and 5 (very often). Participants’ total scores are obtained by adding the score of each answer (ranging from 9 to 45). Higher scores typically indicate a higher level of IGD symptom-severity and greater incidence of problems related to gaming behaviors. For the Spanish adaptation, three experts evaluated (through a table of test specifications) each of the translations and psychological adaptations of the nine items. High inter-rater reliability was recorded throughout the process (> 0.9) on all the items [71]. The set of items is shown in Table 1, however the whole questionnaire including instructions and response scale can be found in Appendix A. In addition, initial piloting was carried out on a sample of 30 participants, providing adequate indicators of reliability and content and internal validity, and not reporting any comprehension or reading problems. The pilot participants were not included in the final sample.

Cuestionario de Experiencias Relacionadas con el teléfono móvil (CERM [Mobile Phone-Related Experiences Questionnaire] [72]). This instrument has 10 items that evaluate two factors: (i) conflicts related to mobile phone abuse and (ii) problems due to the emotional and communicational use of the mobile phone. The items of this instrument are rated on a four-point Likert scale, ranging from 1 (hardly ever) to 4 (almost always). The CERM has been previously shown to have adequate indicators of reliability and validity in Spanish adolescents. In the present sample, the Cronbach’s alpha reliability coefficient for the CERM was 0.78 and the Omega coefficient was 0.79.

Online Gambling Disorder Questionnaire (OGD-Q) [14]. This instrument was designed to evaluate online gambling disorder using a total of 11 items that are rated on a five-point Likert scale ranging from 1 (never) to 5 (every day). The total OGD-Q score varies between 11 and 55, with higher scores indicating higher levels of online disordered gambling. The questionnaire has been validated for a Spanish sample of adolescents and presents adequate indicators of reliability and validity. In the present sample, the Cronbach’s alpha reliability coefficient for the OGD-Q was 0.91 and Omega coefficient was 0.92.

Spanish version of the KIDSCREEN-27 [53]. This instrument assesses HRQoL in children and adolescents between ages 8 and 18 years. This version assesses five dimensions by using 27 items: physical well-being, psychological well-being, autonomy and relationship with parents, peers and social support, and school environment. The development of the KIDCREEN was based on the probabilistic partial credit model (PCM) which pertains to the family of Rasch models. PCM explains the actual behavior of the responders in the testing situation by the estimated person parameter and the location of the item-answers-category-thresholds. The PCM assumes all items of a scale to be the indicators of a single unidimensional latent trait [53]. For the KIDSCREEN-27, the mean scores varied around 50 (SD = 10) due to T-value standardization. There are standardized data for the Spanish infant-juvenile population. The reliability of the each dimension was as follows: physical well-being (α= 0.86; ω = 0.86); psychological well-being (α= 0.84; ω = 0.84), autonomy and relationship with parents (α= 0.84; ω = 0.84), peers and social support (α= 0.88; ω = 0.88), and school environment (α= 0.80; ω = 0.80). Due to the nature of the KIDSCREEN (designed for children and adolescents), responses from participants over 18 years were not considered in the analyses of this instrument in the present study.

### 2.3. Procedure

The battery of questionnaires was applied in an online format utilizing Survey Monkey©. Participants completed the questionnaires in the different computer technology classrooms coordinated by the guidance departments of each center, and under the supervision of the classroom tutor. At the outset of the study participants were advised to answer all questions truthfully and to not stop at any particular question for a long time. The overall average time needed to complete the survey ranged between 15 and 25 min, depending on students’ age and reading ability.

### 2.4. Ethical Considerations

The study was conducted with the informed consent of the participants and the directors of the schools. Through the official communication channels with the families, the schools sent a consent form that informed either the legal tutors or the students themselves (if they were at least 18 years old) about the purpose of the study and its characteristics, its promoters, and their right not to participate without penalties. Those parents/tutors who did not wish to allow participation returned the signed consent. This occurred in less than 1% of the sample. The study was approved by the Research Ethics Committee of the research from Universidad Internacional de la Rioja (UNIR) (PI:008/2019). There were no exclusion criteria, except for the refusal to participate by the legal guardians or by the students themselves.

### 2.5. Data Analysis

Statistical analyses were carried out using the Statistical Package for the Social Sciences (SPSS) [73] program, the R software, the psych package [74], the Lavaan package [75], and the equaltestMI [76]. Firstly, regarding internal validity, an analysis of the psychometric properties of each item was performed, indicating the arithmetic mean, standard deviation, item-total correlation, percentage of positive responses to each item, and the factorial loadings of each item (see Table 1). The multiple criterion for the selection of items without technical deficiencies was that none of them could fail to meet two of the following three statistical indices: (i) a mean between 1.5 and 2.5; (ii) a standard deviation equal to or greater than 1; and (iii) an item-total correlation equal to or greater than 0.35.

To ensure better comparability between the present study and the original IGDS9-SF study, the structure of the IGDS9-SF was initially examined with Exploratory Factor Analysis (EFA) of the items, following the verification of the assumptions (Kaiser-Meyer-Olkin index and Bartlett sphericity test). The factor extraction method used was Principal Axis Factoring with Oblimin rotation. Confirmatory Factor Analysis (CFA) was then performed using Weighted Least Squares Median adjusted method (WLSM). Following the recommendations of Hu and Bentler [77], goodness of fit was assessed using the chi-squared statistic, the comparative fit index (CFI), Tucker-Lewis Index (TLI), the root mean square error of approximation (RMSEA), and the standardized root mean square residual (SRMR). In general, CFI and TLI values of 0.95 or higher reflect a good fit while RMSEA values between 0.06 and 0.08 indicate acceptable fit. Finally, SRMR values lower than 0.08 indicate adequate fit [78]. The hypothesized model was unidimensional, in which all nine items would load on the same latent factor. To determine the internal consistency of the instruments employed, the Cronbach’s alpha, McDonald’s Omega, greatest Lower Bound (GBL), Gutmann’s λ6, Average Variance Explained (AVE) and Composite Reliability (CR) coefficients were estimated. To calculate Measurement Invariance (MI) the sample was split by gender (males and females) and age (under 18 years old and 18 years old or older). MI across age and gender was evaluated through the following steps: (a) testing for the invariance of number of factors (configural invariance); (b) testing for the equality of factor loadings (weak or metric invariance); and (c) testing for the equality of indicator intercepts (strong or scalar invariance). Given that chi-square is sensitive to sample size and non-normality conditions, it was assumed that the model is invariant if the ΔCFI is not above 0.01 [79].

Finally, the following analyses were performed in relation to the secondary objectives of the study: (i) analysis of frequencies and central tendency and dispersion measurements of the study variables; (ii) t-test for independent samples (or failing that, Welch’s test); (iii) calculation of the effect size with Cohen’s d or Hedges’ g, as appropriate; (iv) Pearson correlations; (v) analysis of variance with post-hoc Games-Howell comparisons; and (vi) Mann-Whitney U-test for independent samples. A value of less than *p*=.05 was considered significant.

To obtain the prevalence rate of IGD in the past 12 months, the indications of the APA [21] and Pontes et al. [67] were followed (i.e., endorsing five or more items in classifying individuals with IGD). To establish item endorsement, the items of the IGDS9-SF were dichotomized so that response categories as 4 (often) and 5 (very often) were used to classify the item as a problem (i.e., endorsement of the specific criterion). The remainder of the responses were classified as ‘no problem’ (i.e., no endorsement of the specific criterion). In addition, participants who endorsed four items were considered ‘at-risk’ of IGD while participants’ preferred video game genre was classified in the following categories: action/shooter (FPS (First-Person Shooter), TPS (Three-Person Shooter), etc.), strategy (4x, RTS (Real-Time Strategy), etc.), role-playing (ARPG (Action Role-Playing Games), JRPG (Japanese Role-Playing Game), RPG (Role-Playing Game), etc.), fighting, MOBA (Multiplayer Online Battle Arena), MMORPG (Massive Multiplayer Online Role-Playing Game), simulators, letters, sports, musical, and casual.

## 3. Results

The scores of Items 1, 4, 6, 7, and 9 showed significant differences as a function of gender, with higher scores in males than in females (*p* < 0.05). The rest of the items did not present significant gender differences. The effect sizes were small in most cases (*d* < 0.3), except for Item 1 (*d* = 0.31). Additionally, in relation to the differences according to class year, significant differences were only found on Item 6 (*p* < 0.05) between the students of first basic VT and first middle degree, with a small effect size (*d* < 0.3). No significant differences were observed in the remaining items for the class year variable.

### 3.1. Evidence of Validity of the IGDS9-SF Scores

Table 1 shows the different psychometric indicators for each of the IGDS9-SF items, namely the mean, standard deviation, item-total correlation, percentage of positive response on each item, and factor loadings for each item. At the psychometric level, the scores obtained revealed problems in the mean and standard deviation of all the items, although the item-total correlations in all the items were satisfactory. Items with at least one positive value ranged from 14.8% for Item 9 to 49% for Item 1.

With regards to the EFA results, the data of the Kaiser-Meyer-Olkin index and the Bartlett sphericity test produced values of 0.909 and χ^2^ = 1629.36, *p* < 0.001. The correlation matrix between the items was appropriate for the EFA. The results further indicated a single latent factor explaining 47.49% of the total sample variance. Regarding the CFA, the hypothesized unidimensional model yielded adequate fit indices: χ^2^ (27, n = 532) = 9.908., *p* < 0.001, RMSEA = 0.019 (95% CI [0.000, 0.026], CFI = 0.995, NNFI = 0.993, and SRMR = 0.035. The standardized factor loadings (see Table 1) were statistically significant and notable in all items, ranging from 0.50 to 0.74. The Cronbach’s alpha was 0.85 and Omega coefficients for the IGDS9-SF were both 0.85 (IC: [0.83, 0.87]). The Greatest Lower Bound was 0.88 and Gutmann’s λ was 0.85. Finally, the average variance extracted was 0.5 and the Composite Reliability was 0.88.

### 3.2. Measurement Invariance

To evaluate the generalizability of the model across males and females, participants under 18 years old and 18 years old or older, two multi-group CFAs were performed. For each analysis, an unconstrained model with factor loadings free to vary between subgroups was compared with a more constrained model, in which the factor loadings were held constant across subgroups. Before conducting multi-group analyses, separate CFAs were performed for gender and age subgroups. Regarding gender, the model for females offered a lower fit in general than that of males or the overall model. However, the indicators still presented adequate threshold and the MI analyses were performed. The MI of the single-factor solution was supported at the configural and metric levels. However, the increase in the CFI and RMSEA prevented testing the model any further for gender. Regarding age, both subgroups showed a good fit for the data, in the case of age the MI supported the structure of the single-factor solution across all three levels (configural, metric, and scalar). The results obtained for the different models are displayed in Table 2.

### 3.3. Convergent Validity

In relation to convergent validity, the Pearson’s bivariate correlation carried out between the total IGDS9-SF scores and the OGD-Q scores had a value of r = 0.440, *p* < 0.001 (*n* = 101); with the CERM, it was *r* = 0.553, *p* = 0.001. Additionally, IGDS9-SF correlated with the five dimensions of the KIDSCREEN-27 as follows: physical well-being (*r* = −0.164, *p* = 0.001), psychological well-being (*r* = −0.315, *p* = 0.001), autonomy and relationship with parents (*r* = −0.167, *p* = 0.001), peers and social support (*r* = −0.257, *p* = 0.001), and school environment (*r* = −0.176, *p* = 0.001).

It was also found that gamers who preferably played the MOBA-, RPG-, or MMORPG-type game genres reported higher scores on the IGDS9-SF (15.21 ± 6.14) than those who played other genres (FPS, action/platforms, musical, sports simulators, fighting, or casual) (13.81 ± 4.55) (*t* = 2.679, *p* < 0.008, *d* = 0.26). Additionally, in response to the question ‘*I am addicted to video games’*, 65 answered ‘yes’, and 473 replied ‘no’. Those who self-reported that they were addicted had a significantly higher mean score on the IGDS9-SF (19.19 ± 8.30) compared to those who did not (13.43 ± 4.73) (*t* = 8.233, *p* < 0.001, d = 0.85).

### 3.4. Prevalence and Psychological Involvement of Internet Gaming Disorder

Following the diagnostic approach suggested by Pontes et al., [61], the participants who were classified with IGD (i.e., endorsing at least five of the criteria within the last 12 months) accounted for 1.9% of the sample of gamers (*n* = 10; see Table 3) and almost 1% of the total study sample. Of these 10 participants, nine were male and one was female. In addition, 1.9% endorsed four diagnostic criteria (*n* = 10) and were classed as ‘at-risk’ of developing IGD. It should also be noted that 76.2% (*n* = 410) did not endorse any diagnostic criteria.

Table 4 shows the psychological involvement of players with IGD (those with five or more symptoms) compared to those with four or fewer symptoms in relation to HRQoL. The loss of physical (*p* = 0.011) and psychological well-being (*p* = 0.018) is especially notable. There was also a loss of autonomy and relationship with parents (*p* = 0.047) and worse school environment (*p* = 0.038). There were no differences for the dimension of peers and social support (*p* = 0.080). The effect sizes for all contrasts were greater than *g* > 0.42.

## 4. Discussion

The IGDS9-SF is a sound psychometric test that assesses IGD, and it is one of the most frequently used instruments as it had the greatest number of adaptations to different languages and cultural contexts [56]. In the present study, the development of the Spanish IGDS9-SF was carried out through a rigorous conceptual and methodological procedure that followed conventional international standards [71]. Appropriate indicators of validity and reliability were obtained in a sample of adolescents and young people. Factor analysis confirmed a single-factor solution with adequate goodness of fit, the item-total correlations were also high, and the factor loadings of all the items were satisfactory. Furthermore, the unidimensional factor model was found to be gender and age invariant across the metric level, which is considered a prerequisite for meaningful cross-group comparisons [80]. The present study adopted a similar procedure to studies conducted for different constructs, such as nomophobia [81,82,83], online gambling disorder [14], and previous validations of the IGDS9-SF in other languages such as Portuguese [67], Slovenian [61], and Italian [62], among others.

The prevalence rate of IGD in the present sample was 1.9%, which is similar to the 2.6% found in another Spanish sample by Fuster et al. [37] but noticeably lower than that of 8.3% reported by Buiza-Aguado et al. [36]. These discrepancies may be due to the use of different psychometric tools in assessing IGD. In other studies that have utilized the IGDS9-SF, the prevalence rates of disordered gaming were reported to be between 3% to 5% [59,61]. The results of the present study fall within the prevalence rate range reported by other international studies [31]. Finally, it also corroborates the fact that males are more frequently classified with IGD than females [31,37,38].

In relation to other validity indicators, the present study sought to evaluate the relationship between the IGDS9-SF and instruments assessing conceptually similar psychological problems such as the CERM [72] and the OGD-Q [14]. The results obtained indicated high correlations between these constructs, suggesting a convergent relationship with other relevant problems related to maladaptive use of technology and the internet such as problematic smartphone use and online gambling disorder. In addition, indicators relating to the relationship between IGDS9-SF and HRQoL allowed the examination of five key dimensions in adolescence (i.e., physical well-being; psychological well-being; autonomy and relationship with parents; peers and social support; and the school environment). Inverse and significant relationships were found between all five aforementioned dimensions, which means that higher level problems with online gaming associate with poorer self-reported quality of life. It is especially interesting to compare participants categorized as gamers with IGD or who are at risk with those who are not, because there was a significant decrease in quality of life scores in the former. Overall, the effects sizes of these correlations were high (i.e., most were greater than 0.80). These results are consistent with those of other studies using the 20-item Internet Gaming Disorder Test (IGD-20 Test [24]) and the KIDSCREEN-27 [31]. This finding also has a theoretical-conceptual relationship with the components model of addiction by Griffiths [23], which highlights the importance relating to the negative effects of the symptoms of addiction. In addition, it also supports the notion of other more general conceptualizations of problematic use of the internet [8], in which problems are related to poorer social and personal functioning, as well as to compulsive use and negative consequences.

### Limitations and Future Lines of Research

The study conducted presents with several potential limitations worth discussing. Firstly, the IGDS9-SF is a self-report psychometric tool, so the potential for confounding effects stemming from response biases and social desirability by the adolescents and young people who completed it cannot be completely ruled out. This could be improved in the future by developing complementary measures combining behavioral tracking data (e.g., actual time spent playing and in-game preferences) to enhance self-report data. Secondly, the sample recruited was not randomly selected. However, the sample size of the present study was significantly large, especially in the context of a psychometric study. Nevertheless, caution is suggested when extrapolating the prevalence rates reported in the present study and considering them as a first approximation to the problem. Thirdly, although the diagnostic approach of the APA [21] and the recommendation of the original authors of the IGDS9-SF were followed when classifying disordered gamers, the authors have sought to establish a less conservative diagnostic approach which requires further discussion and analyses. Fourthly, whereas adequate indicators of validity and reliability were obtained, other important measures such as test-retest were not considered due to the imperatives of fieldwork. Fifthly, the KIDSCREEN-27 is a tool designed to evaluate HRQoL in the infantile-juvenile population, and the sample here included some participants over age 18 years. Thus, participants over 18 years were not considered in the analyses related to this construct, which reduced the sample size. It would be of interest for future research to use developmentally specific quality of life assessment tools for those over 18 years. It would also be fruitful to explore in future research other processes that favor the diagnostic accuracy of this scale, such as Receiver Operating Characteristic (ROC) curves (as demonstrated by Severo et al. [68] and Monacis et al. [62]), and other diagnostic elements, such as interviewing or complementary measures, should be used in the future in order to establish a robust clinically-driven gold standard diagnosis. Finally, according to the data obtained, the possible relationship between IGD and online gambling disorder with the advent of gambling-type elements in video games (e.g., loot boxes) should be noted for future lines of research to be explored. Legally, loot boxes are not considered online gambling, but at a psychosocial level, they meet the characteristics to be defined as a type of gambling [84].

## 5. Conclusions

The present study corroborates the psychometric properties of the scores obtained on the IGDS9-SF. In addition, preliminary data on the prevalence of the disordered gaming were obtained, which are useful for knowledge of an emerging global health challenge. The findings reported here will be particularly useful to pediatric and psychological care units, as well as for those in charge of school orientation at schools. All the above is also of special interest to parents, because education and parental supervision can play a very important role in the prevention of these problems associated with the internet, and in particular, internet gaming.

## Figures and Tables

**Table 1 ijerph-17-01562-t001:** Means, standard deviations, item-total correlation, positive response percentage, and factorial loads of the Internet Gaming Disorder Scale–Short Form (IGDS9-SF) items (*n* = 9).

IGDS9-SF Items	M	SD	IT	%+	CFE
1. ¿Te sientes preocupado por tu comportamiento con el juego? (Algunos ejemplos: ¿Piensas en exceso cuando no estás jugando o anticipas en exceso a la próxima sesión de juego?, ¿Crees que el juego se ha convertido en la actividad dominante en tu vida diaria?)	1.74	0.95	0.53	48.0	0.58
2. ¿Sientes irritabilidad, ansiedad o incluso tristeza cuando intentas reducir o detener tu actividad de juego?	1.45	0.79	0.64	32.5	0.71
3. ¿Sientes la necesidad de pasar cada vez más tiempo jugando para lograr satisfacción o placer?	1.57	0.97	0.67	34.3	0.74
4. ¿Fallas sistemáticamente al intentar controlar o terminar tu actividad de juego?	1.64	0.89	0.63	43.2	0.69
5. ¿Has perdido intereses en aficiones anteriores y otras actividades de entretenimiento como resultado de tu compromiso con el juego?	1.56	0.93	0.61	34.3	0.66
6. ¿Has continuado jugando a pesar de saber que te estaba causando problemas con otras personas? (pareja, amistad o familia)	1.45	0.88	0.65	27.2	0.71
7. ¿Has engañado a alguno de tus familiares, terapeutas o amigos sobre el tiempo que pasas jugando?	1.57	0.99	0.47	33.6	0.51
8. ¿Juegas para escapar temporalmente o aliviar un estado de ánimo negativo (por ejemplo, desesperanza, tristeza, culpa o ansiedad)?	1.94	1.20	0.47	49.0	0.50
9. ¿Has comprometido o perdido una relación importante, un trabajo o una oportunidad educativa debido a tu actividad de juego?	1.23	0.64	0.58	14.8	0.63

Note: For original items see Pontes and Griffiths [57]; M = Arithmetic mean; SD = Standard deviation; IT = corrected item-total correlation; %+ Percentage that has responded positively (at least once); CFE = Standardized factorial loads.

**Table 2 ijerph-17-01562-t002:** Invariance analyses across gender and age.

Model	χ^2^	df	Com. Md	ΔSB χ^2^	Δdf	*p*	CFI	ΔCFI	RMSEA	ΔRMSEA	SRMR
1.Overall model	9.91	27	--	--	--	--	0.995	--	0.019	--	0.035
**Gender**											
2. Men Model	9.08	27	--	--	--	--	0.994	--	0.02	--	0.037
3. Women Model	12.95	27	--	--	--	--	0.984	--	0.035	--	0.072
4. Configural Model	22.03	54	--	--	--	--	0.992	--	0.024	--	0.044
5. Metric Model	39.05	62	4–5	9.57	8	0.296	0.989	−0.003	0.026	0.002	0.058
6. Scalar Model	57.11	70	5–6	26.81	8	>0.001	0.976	−0.013	0.036	0.010	0.062
**Age**											
7. ≤17 Model	5.38	27	--	--	--	--	0.998	--	0.012	--	0.035
8.≥18 Model	11.93	27	--	--	--	--	0.988	--	0.029	--	0.049
9. Configural Model	17.33	54	--	--	--	--	0.992	--	0.023	--	0.043
10. Metric Model	28.85	62	9–10	11.99	8	0.151	0.988	−0.004	0.026	0.003	0.056
11. Scalar Model	37.40	70	10–11	14.60	8	0.464	0.983	−0.005	0.029	0.003	0.058

Note: *n* for Men’s model = 419; *n* for women’s model = 113; *n* for ≤ 17’s Model = 296; *n* for ≥ 18’s model = 236; χ^2^ = Chi-Squared; df = Degrees of freedom; Comp. Md = Compared models; Δ SBχ^2^ = differences in Satorra-Bentler Scaled Chi-Squared; Δdf = difference in number of degrees of freedom; p = significance value for the Scaled Chi-Squared Difference Test; CFI: Comparative Fit Index; ΔCFI = differences in Comparative Fit Index; RMSEA = Root Mean Square Error of Approximation; SRMR = Standardized Root Mean Square Residual;.

**Table 3 ijerph-17-01562-t003:** Number of participants who meet between 1 and 9 of the Internet Gaming Disorder criteria (adapted from Pontes, et al. [61]).

Number of Criteria Endorsed	Number of Participants	Total % of the Sample (*n* = 1064)	Total % of Online Gamers (*n* = 535)
1	71	6.67	13.2
2	27	2.54	5
3	10	0.94	1.9
4	10	0.94	1.9
5	1	0.09	0.2
6	1	0.09	0.2
7	2	0.19	0.4
8	2	0.19	0.4
9	4	0.38	0.8

**Table 4 ijerph-17-01562-t004:** Comparison between gamers with Internet Gaming Disorder (IGD) (endorsing five or more criteria) versus those endorsing four criteria or fewer with respect to the five dimensions of the KIDSCREEN-27.

Instrument	*n* (< 5 symptoms)	*M* ± *SD* (< 5 symptoms)	*n* (≥ 5 symptoms)	*M* ± *SD* (≥ 5 symptoms)	Mann-Whitney U (*p*)	Effect Size (*Hedges’ g*)
KD Phy-Wb	341	45.40 ± 11.39	7	33.92 ± 10.51	2.529 (.011)	1.00
KD Psy-Wb	341	46.70 ± 9.47	7	38.00 ± 8.88	2.370 (.018)	0.92
KD A and Pr	341	48.51 ± 10.91	7	39.75 ± 21.70	1.749 (.080)	0.78
KD SS and P	341	50.97 ± 11.31	7	37.81 ± 20.06	1.988 (.047)	1.14
KD SE	341	46.19 ± 9.10	7	42.31 ± 13.26	2.076 (.038)	0.42

Note: n (< 5 symptoms): participants with fewer than four problem criteria/items; n (≥ 5 symptoms): participants with four or more problem criteria/items; KD Phy-Wb = Physical well-being; KD Psy-Wb = Psychological well-being; KD A and Pr = Autonomy and relations with parents; KD SS and P = Social Support and Peers; KD; SE = School Environment; M = arithmetic mean; SD = standard deviation.

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
