# Peer review of "Spanish Validation of the Internet Gaming Disorder Scale–Short Form (IGDS9-SF): Prevalence and Relationship with Online Gambling and Quality of Life"

_ijerph, 2020, doi:10.3390/ijerph17051562_

Round 1

Reviewer 1 Report

This is a useful and very well written article. It will be useful for for those researching internet addiction, and especially for those in Spanish -speaking countries.

I only have a couple of very minor editing suggestions:

line 68, after 'addiction' I think 'and' or 'with' is missing

line 90 and line 93 I think 'about' can be deleted as the percentages are quite precise.

Author Response

REVIEWER COMMENTS FOR THE AUTHOR Reviewer #1:

Reviewer #1 This is a useful and very well written article. It will be useful for those researching internet addiction, and especially for those in Spanish -speaking countries.

I only have a couple of very minor editing suggestions:

line 68, after 'addiction' I think 'and' or 'with' is missing

line 90 and line 93 I think 'about' can be deleted as the percentages are quite precise.

Authors --> Thank you very much for your general assessment and for your suggestions for improving the manuscript. The authors have included these two changes in the manuscript.

Reviewer 2 Report

Thank you for the opportunity to review the manuscript entitled, "Spanish Validation of the Internet Gaming Disorder Scale–Short Form (IGDS9-SF): Prevalence and relationship with online gambling and quality of life”.

I believe this study investigated a topic relevant to the readers of “IJERPH”. The use of Information Technology and Communications (ICT) is an essential aspect of modern societies. Access to such tools is increasingly easy and their use is not problem-free. Video games represent a very popular leisure activity in many countries, as Spain. The Internet Gaming Disorder (IGD) has recently been introduced in the nosological manuals and is considered an addictive behavior. Over the last years, research on video game addiction has sharply increased and several instruments have been recently developed to measure video game addiction. I think that is very important to develop standardized instruments based on the criteria proposed by the DSM-5 to measure IGD.

This paper is well written and follows well accepted standards of academic writing. However, minor revisions may prove beneficial.

Introduction:        

The introduction is very brief and simply, not analyze in detail the relationship between the IGD and Health-Related Quality of Life (HRQoL). It is an objetive very important in this study.

In relation to convergent validity, need be addressed in the introduction section. Why this instruments: OGD-Q, CERM and KIDSCREEN-27?

Spanish is the fourth most spoken language in the world. It is a mistake?

Data analysis and results:

In first place, is need the Composite Reliability (CR) and Average Variance Extracted (AVE) of the IGDS9-SF, CERM, OGD-Q and KIDSCREEN-27.

On the other side, the authors should randomly divide the original sample in two parts, in order to have two independent subsamples. The first sample is subjected to an exploratory factor analysis and the second sample is subjected to a confirmatory factor analysis.

In third place, the assumptions of linearity and normal distribution of all the variables observed in the model should be met to be able to use the maximum likelihood method (Jöreskog & Sörbom, 1996). Also, is need a multi-group analysis to determine whether the models are invariant by gender and class year.

Author Response

REVIEWER COMMENTS FOR THE AUTHOR - Reviewer #2:

Reviewer #2 Thank you for the opportunity to review the manuscript entitled, "Spanish Validation of the Internet Gaming Disorder Scale–Short Form (IGDS9-SF): Prevalence and relationship with online gambling and quality of life”.

I believe this study investigated a topic relevant to the readers of “IJERPH”. The use of Information Technology and Communications (ICT) is an essential aspect of modern societies. Access to such tools is increasingly easy and their use is not problem-free. Video games represent a very popular leisure activity in many countries, as Spain. The Internet Gaming Disorder (IGD) has recently been introduced in the nosological manuals and is considered an addictive behavior. Over the last years, research on video game addiction has sharply increased and several instruments have been recently developed to measure video game addiction. I think that is very important to develop standardized instruments based on the criteria proposed by the DSM-5 to measure IGD.

This paper is well written and follows well accepted standards of academic writing. However, minor revisions may prove beneficial.

Authors --> Thank you very much for your general assessment and for your suggestions for improving the manuscript.

Reviewer #2 Introduction:       

The introduction is very brief and simply, not analyze in detail the relationship between the IGD and Health-Related Quality of Life (HRQoL). It is an objetive very important in this study.

Authors --> The authors have incorporated a paragraph to discuss the HRQoL construct and its relationship to IGD in the revised Introduction. You can find the new content on lines 116-129 in the new version of the manuscript (R1).

Reviewer #2 In relation to convergent validity, need be addressed in the introduction section. Why this instruments: OGD-Q, CERM and KIDSCREEN-27?

Authors --> The authors have now incorporated into the Introduction further information about the constructs/questionnaires used to obtain convergent validity. You can find the new content in lines 104-128 of the new version of the manuscript (R1).

Reviewer #2 Spanish is the fourth most spoken language in the world. It is a mistake?

Authors -->According to the website consulted (reference 64, "https://www.ethnologue.com/guides/ethnologue200") Spanish is the fourth most spoken language in the world. The first is English, the second is Mandarin and the third is Hindi. The data vary according to the sources consulted. According to the annual report of the Cervantes Institute, Spanish it is the third. We have changed that data although both sources are reliable.

Reviewer #2 Data analysis and results:

In first place, is need the Composite Reliability (CR) and Average Variance Extracted (AVE) of the IGDS9-SF, CERM, OGD-Q and KIDSCREEN-27.

Authors --> The authors have incorporated the standard indicators of reliability (i.e., Cronbach's alpha and the Omega coefficient). Both are satisfactory for all dimensions used in this study. The CR and the AVE have been included specifically for the IGDS9-SF, which is the scale that was validated. You can find the new content in lines 311-312 of the new version of the manuscript (R1).

Reviewer #2 On the other side, the authors should randomly divide the original sample in two parts, in order to have two independent subsamples. The first sample is subjected to an exploratory factor analysis and the second sample is subjected to a confirmatory factor analysis.

Authors --> We agree with you. The most appropriate procedure is the one you indicate. This study presents an important limitation, as a first approximation of the validated Spanish IGDS9-SF, and that is that the sample slightly exceeds 500 participants. If we divide it to perform an AFE first and then an AFC, the sample for confirmatory analysis becomes very small (<300). Authors such as Brown (2006) suggest not to make this division.

Comrey and Lee (1992, p. 217) suggested that: "the adequacy of the sample size could be evaluated with the following scale: 50 -very poor; 100 -poor; 200 -acceptable; 300 good, 500-very good, 1000 or more-excellent." Although it is recommended to reach 500 or more cases, whenever possible (MacCallum et al. 1999). 

Brown, T. A. (2006). Confirmatory factor analysis for applied research. New York: Guilford Press.

Comrey, A. L. & Lee, H. B. (1992). A first course in factor analysis. Hillsdale, NJ: Erlbaum

MacCallum, R. C., Widaman, K. F., Zhang, S. y Hong, S. (1999). Sample size in factor analysis. Psychological Methods, 4, 84-99.

Reviewer #2 In third place, the assumptions of linearity and normal distribution of all the variables observed in the model should be met to be able to use the maximum likelihood method (Jöreskog & Sörbom, 1996).

Authors --> Thank you for this comment. The authors followed the procedure of other adaptations of the IGDS9-SF to replicate the statistical analyses and so we opted for the Maximum Likelihood Robust. However, what you suggest is correct and we have modified it (in the revised Data analysis section and in the revised Results section). The model that is now used uses a robust categorical estimator, specifically the Weighted Least Squares Mean adjusted method (WLSM).

Reviewer #2 Also, is need a multi-group analysis to determine whether the models are invariant by gender and class year.

Authors --> The authors have now incorporated this suggestion in the revised manuscript. We believe that this change implied a major (not minor) revision. Important changes have been included in the section on Data analysis (lines 241-268), in the Results (299; 306-332), and the Discussion (373-385).

Reviewer 3 Report

The article submitted for a review: Spanish Validation of the Internet Gaming Disorder Scale – Short Form (IGDS9-SF): Prevalence and relationship with online gambling and quality of life
concerns the important scientific purpose of establishing psychometric properties of the Spanish version of the IGDS9-SF scale.
The article was correctly formatted, divided into parts in accordance with the requirements of the magazine. It is written in an interesting way and has appropriately selected, widely analyzed and current literature. A psychometric analysis of the properties of the Internet Gaming Disorder Scale-Short Form (IGDS9-SF) was conducted correctly. The article provides important scientific support for the transmission of research results to the practice of the process of diagnosis and therapy of Internet addiction among the Spanish-speaking world population.
It is worth supporting the diligence of this publication before publishing by supplementing the article with
• clearly separated from the rest of the text possible restrictions of Author’s work (in the current form it is part of the discussion)
• record of analyzes carried out relative to the tool's relevance, including theoretical, diagnostic and prognostic validity

The text is worth publishing, which I recommend.

Author Response

REVIEWER COMMENTS FOR THE AUTHOR - Reviewer #3:

Reviewer #3  The article submitted for a review: Spanish Validation of the Internet Gaming Disorder Scale – Short Form (IGDS9-SF): Prevalence and relationship with online gambling and quality of life

concerns the important scientific purpose of establishing psychometric properties of the Spanish version of the IGDS9-SF scale.

The article was correctly formatted, divided into parts in accordance with the requirements of the magazine. It is written in an interesting way and has appropriately selected, widely analyzed and current literature. A psychometric analysis of the properties of the Internet Gaming Disorder Scale-Short Form (IGDS9-SF) was conducted correctly. The article provides important scientific support for the transmission of research results to the practice of the process of diagnosis and therapy of Internet addiction among the Spanish-speaking world population.

It is worth supporting the diligence of this publication before publishing by supplementing the article with

  • clearly separated from the rest of the text possible restrictions of Author’s work (in the current form it is part of the discussion)
  • record of analyzes carried out relative to the tool's relevance, including theoretical, diagnostic and prognostic validity

The text is worth publishing, which I recommend.

Authors --> Thank you very much for your general assessment and for your suggestions for improving the manuscript. The authors have made the changes in the manuscript. We have added a subsection with the Limitations and another with the Conclusions.

Regarding your second comment, we are sorry to say that we do not understand what you mean.